Prevalence of type 2 diabetes mellitus and impaired fasting glucose, and their associated lifestyle factors among teachers in the CLUSTer cohort

Ng Yit Han 1
Moy Foong Ming moyfm@ummc.edu.my 1 2
Hairi Noran Naqiah 1 2
Bulgiba Awang 3
1 Department of Social and Preventive Medicine, Faculty of Medicine, Universiti Malaya , Kuala Lumpur , Malaysia
2 Centre for Epidemiology and Evidence-Based Practice, Universiti Malaya , Kuala Lumpur , Malaysia
3 Academy of Sciences Malaysia , Kuala Lumpur , Malaysia
Foti Daniela
Electronic publication date: 2024 Jan 22
Publication date: 2024
Volume: 12
Electronic Location ID: e16778
Received 2023 Jul 28; Accepted 2023 Dec 18
Copyright: ©2024 Ng et al.
Copyright year: 2024
Copyright holder: Ng et al.
License: This is an open access article distributed under the terms of the Creative Commons Attribution License, which permits unrestricted use, distribution, reproduction and adaptation in any medium and for any purpose provided that it is properly attributed. For attribution, the original author(s), title, publication source (PeerJ) and either DOI or URL of the article must be cited.
License URL: https://creativecommons.org/licenses/by/4.0/

Keywords: Type-2 Diabetes, Impaired fasting glucose, Teachers, Lifestyles, Cohort

Funding: Ministry of Science, Technology, and Innovation (MOSTI) Malaysia FP 080-2020 Universiti Malaya Faculty of Medicine’s Postgraduate Scholarship Fund This work was supported by the Ministry of Science, Technology, and Innovation (MOSTI) Malaysia via a fundamental research grant scheme (FP 080-2020) and the Universiti Malaya Faculty of Medicine’s Postgraduate Scholarship Fund. The funders had no role in study design, data collection and analysis, decision to publish, or preparation of the manuscript.

==============================
Background

Teachers are responsible for educating future generations and therefore play an important role in a country’s education system. Teachers constitute about 2.6% of all employees in Malaysia, making it one of the largest workforces in the country. While health and well-being are crucial to ensuring teachers’ work performance, reports on non-communicable diseases such as type 2 diabetes mellitus (T2DM) among Malaysian teachers are scarce. Hence, this study focused on the prevalence of T2DM, undiagnosed diabetes mellitus (DM), impaired fasting glucose (IFG), and underlying lifestyle factors associated with these outcomes among Malaysian teachers.

Methods

This is a cross-sectional study from the CLUSTer cohort. There were 14144 teachers from the Peninsular Malaysia included in this study. The teachers’ sociodemographic and lifestyle characteristics were described using a weighted complex analysis. A matched age group comparison was carried out between teachers and the Malaysian general population on T2DM, undiagnosed DM, and IFG status. Next, the researchers examined the association of lifestyle factors with T2DM and IFG using multivariable logistic regression.

Results

The prevalence of T2DM, undiagnosed DM, and IFG among the Malaysian teachers were 4.1%, 5.1%, and 5.6%, respectively. The proportions of teachers with T2DM (both diagnosed and undiagnosed) and the IFG increased linearly with age. Teachers had a lower weighted prevalence of T2DM (known and undiagnosed) than the general population. However, teachers were more inclined to have IFG than the general population, particularly those aged 45 years and older. Among all lifestyle indicators, only waist circumference (aOR: 1.14, 95% CI: 1.08, 1.20) was found to be associated with T2DM, whereas waist circumference (aOR: 1.10, 95% CI: 1.05, 1.15) and physical activity [moderately active = (aOR: 0.71, 95% CI: 0.52, 0.98); highly active = (aOR: 0.56, 95% CI: 0.40, 0.80)] were associated with IFG.

Conclusions

Modifiable lifestyle factors such as abdominal obesity and physical activity were associated with T2DM and IFG. Intervention programs targeting these factors could help reduce future treatment costs and increase productivity.

Introduction

Teachers play an important role in shaping a nation’s future and their responsibilities extend beyond the classroom. Along with other education-related professionals, they are tasked with teaching knowledge and values to future generations, ultimately contributing to the achievement of a country’s goals. Teachers made up 2.6% of all employees in Malaysia as of April 2022, making up one of the most significant labour forces in the nation (Department of Statistics Malaysia, 2022; Ministry of Education Malaysia, 2022). Most academic professionals are schoolteachers compared to other educators such as private tutors and lecturers.

Teachers are susceptible to occupational illnesses and injuries such as bone or joints, stress, eyesight, and vocal/throat issues because of the nature of their work (Bhattarai, Bashistha & Acharya, 2019). Thus, researchers tend to concentrate on teachers’ well-being and work-related diseases (Moy et al., 2015; Pau et al., 2022; Tai, Ng & Lim, 2019; Zamri, Moy & Hoe, 2017). While chronic diseases such as type 2 diabetes mellitus (T2DM) can also affect teachers, information on its prevalence is scarce because it is rarely studied among teachers in Malaysia. T2DM is a metabolic disease caused by a failure in glycaemic control, mainly due to either insulin resistance or defective insulin secretion in a person’s body by pancreatic beta-cells (Galicia-Garcia et al., 2020). Consequently, macrovascular and microvascular complications of diabetes affect work performance and productivity (Nanayakkara et al., 2021).

The prevalence of T2DM in the general population of Malaysia has increased from 8.3% to 9.4% between 2015 and 2019 (Institute for Public Health, 2015; Institute for Public Health, 2020). Likewise, the prevalence of undiagnosed T2DM in the general population is slightly higher than that among individuals diagnosed with T2DM (Institute for Public Health, 2015). Similar surveys have also stated that individuals with Impaired Fasting Glucose (IFG) levels have increased from 5.1% to 8.9% over the years (Institute for Public Health, 2020). In such a scenario, there will be an increase in disability adjusted for life years (DALYs) in this country due to the drastic increase in T2DM incidence (Kyu et al., 2018). In addition to Malaysia, an increasing trend in T2DM also poses a significant health threat globally, especially among European Union countries, which also report increasing DALYs (Goodall et al., 2021).

A healthy lifestyle involves several components, such as maintaining healthy diet intake, engaging in regular exercise, managing body weight, refraining from smoking, and controlling alcohol intake. These factors are frequently included in diabetes intervention programs. However, some of these common lifestyles, such as smoking and alcohol consumption, are not relevant for specific occupational groups such as teachers, as they are less likely to be involved in high risk behaviours compared to other professions (Temam et al., 2022). To date, there has been collective evidence of lifestyle factors related to metabolic syndrome or disorders among teachers in other countries. These studies demonstrated that poor weight management, indirect exposure to cigarette smoke, physical inactivity, unhealthy diet (sugar-sweetened beverages), sedentary work style, and persistent stress conditions are risk factors for metabolic syndromes, as well as metabolic disorders such as diabetes and hypertension, among teachers in different countries (Damtie et al., 2021; Jiang et al., 2019; Narayanappa, Manjunath & Kulkarni, 2016; Stern et al., 2019; Zubery, Kimiywe & Martin, 2021). In addition, a study reported sleep health and appropriate sleep duration as additional lifestyle factors associated with metabolic syndrome in school teachers in Malacca State, Malaysia (Lee, Hairi & Moy, 2017). Despite conducting national health surveys every four years, these findings may not accurately reflect the health status of teachers in this country. Consequently, there is still a lack of information on non-communicable diseases among teachers in Malaysia. Furthermore, lifestyle factors, such as sleep duration and sedentary behaviours, along with acute mental health conditions which have been reported to be relevant risk factors, require further exploration.

This study aimed to determine the prevalence of known T2DM, undiagnosed DM, and IFG among teachers in the CLUSTer cohort, followed by comparison with findings from the Malaysia National Health and Morbidity Survey (NHMS). The secondary objective aimed to explore and investigate lifestyle factors associated with T2DM and IFG among teachers.

Methods

CLUSTer cohort

CLUSTer is a cohort study of Malaysian teachers with the primary goal of understanding the clustering of lifestyle risk factors on teachers’ health and well-being, including type 2 diabetes (Moy et al., 2014). Using a multi-staged sampling technique, approximately 14,500 primary and secondary schoolteachers were recruited between 2013 and 2015. They were recruited from 534 schools in six study areas (Penang, Selangor, Terengganu, Malacca, Johor, and Kuala Lumpur) in Peninsular Malaysia.

Study design

This is a cross-sectional study from the CLUSTer cohort. During baseline recruitment, all eligible teachers free from psychiatric illnesses were asked to answer a questionnaire containing demographic information, health status, and lifestyle. The investigators also collected anthropometric parameters such as waist circumference. Teachers’ blood pressure and fasting blood glucose levels were measured using a portable blood pressure device and a laboratory venous glucose test, respectively.

Study variables

The demographic variables were age, gender, ethnicity, education level, marital status, and family history of T2DM. Lifestyle variables included daily fruit and vegetable consumption (servings per day), physical activity measured by the International Physical Activity Questionnaire short version (IPAQ-short), smoking status, alcohol consumption, waist circumference, sitting duration, sleep duration, and acute mental health status (depression, anxiety, and stress rated using the Depression, Anxiety, and Stress Scale 21 questions version (DASS-21)). According to the Ministry of Health guidelines, at least three daily vegetable and two fruit servings are recommended (Ministry of Health Malaysia, 2020). As for physical activity, at least five days of moderate-intensity physical activity (30 min) is required to maintain health (Ministry of Health Malaysia, 2020).

The outcomes of this study included self-reported T2DM, undiagnosed T2DM and IFG status. Teachers’ blood samples were collected and stored in a −80 Celsius freezer before being analysed for fasting blood glucose (FBG) at the University of Malaya Medical Centre laboratory. To make the results comparable with those of the NHMS survey, the authors adopted the outcome definition by the Institute of Public Health 2015 for undiagnosed T2DM and IFG. Participants who previously did not report having T2DM but had FBG laboratory readings ≥ 6.1 mmol/L were considered to have undiagnosed T2DM. Conversely, teachers who were free of T2DM yet their FBG levels exceeded 5.6 mmol/L but were below 6.1 mmol/L were considered to be co-living with IFG (Institute for Public Health, 2015).

The operational definitions of all the variables are described in Table S1.

Statistical analysis

In descriptive analysis, a complex analysis was conducted using a “survey” package in R software (Lumley, 2004). A complex analysis incorporated an assigned weight throughout the data analysis (Lohr, 1999). The complex sample weight was computed based on the multiplication of the two-staged sampling units, known as school and teacher (Fig. S1). This total weight was applied to compute the weighted means with standard deviation for all scaled variables whereas frequencies with weighted percentages were used for all categorical variables. The weight was used to account for the multi-level data, including the unequal probability of selection and non-respondents. A matched age group comparison was performed between teachers and the Malaysian general population on the prevalence of T2DM, undiagnosed DM, and IFG. Briefly, the Malaysian general population in the NHMS survey had approximately equal proportions of males and females. Approximately 60% of the investigated Malaysian population was married and two-thirds of them were employed. However, ethnicity information was not disclosed. Further details of the survey can be assessed from the Institute for Public Health (2015).

Teachers who answered the majority of the survey questionnaires were included in the inferential analysis. A multiple imputation chained equation (MICE) technique was applied to deal with all missing data (Van Buuren et al., 2021). In bivariate and multivariable factor association analyses, univariable and multivariable logistic regressions were performed using the same “survey” package. All non-lifestyle covariates with p value <0.25 were included in the multivariable model, along with all lifestyle factors. The multicollinearity test was used to detect high correlations between variables included in the multivariable regression model. Variables had variance inflation factors below 5.0 suggesting no multicollinearity issue (Heiberger & Holland, 2015). Lifestyle variables with p values <0.05 determined via multivariable analysis were considered significantly associated with T2DM and IFG, respectively. Additional analyses on interaction were conducted, and significant findings were incorporated into the final models. A “ggeffect” package in R (Lüdecke, 2018) was used to illustrate the multivariable regression findings. Finally, additional stratified analyses were conducted, both descriptively and inferentially among male and female teachers.

Ethical clearance

This study obtained ethical approval from University of Malaya Medical Centre Medical Research Ethics Committee (UMMC MREC) (MREC ID: 950.1). Hardcopies of written consent were obtained from all participants.

Results

Teacher’s characteristics, prevalence of T2DM and IFG among the teachers

A total of 14,144 teachers out of the initial sample of 14,427 teachers were included in the analysis after excluding duplicates. The enrolled teachers were around 40 years old, and four-fifths were female. About two-thirds (n = 10773 ,77%-weighted) of the sample were of Malay ethnicity, followed by Chinese (n = 2291, 14%-weighted), Indian (n = 979, 8%-weighted), and other races (n = 101, 1%-weighted). Most of them had at least a bachelor’s degree and above, and at least 80% were married. Approximately half of the teachers reported a family history of T2DM.

Teachers’ lifestyles and daily fruit and vegetable servings were below the recommended serving size, and only 2.3% of teachers had adequate fruit (two) and vegetable (three) servings daily. Similarly, teachers also lacked sleep, particularly on weekdays, where they only slept for an average of six hours compared to seven hours on weekends. Physical activity and sitting duration varied among the teachers. The metabolic equivalent of tasks (METs), in minutes per week, among teachers had a wide range, with a weighted average of 1,344.58, and a dispersion range of 2,908.50. Likewise, the weighted average sitting duration ranged from one to six hours daily. Only 3.7% and 4.0% of teachers were ever smokers and alcohol consumers, respectively. As for weight management, teachers’ documented waist circumferences ranged from 70 to 90 cm, and 52% of them had abdominal obesity, after adjusting for sex. Finally, for acute mental health status, the average depression, anxiety, and stress scores were six, eight, and 10, respectively, but all scores had a standard deviation of six. The high variations in all acute mental health scores indicated that teachers’ mental health status also varied.

The prevalence of known T2DM among teachers was 4.1% (95% CI: 3.7, 4.5),  whereas the prevalence of undiagnosed T2DM was as high as 5.4% (95% CI: 4.9, 6.0). Among the non-diabetic teachers (n = 12,757), about 5.6% (95% CI: 5.1, 6.2) experienced IFG during the baseline survey. The details of the teachers’ characteristics are summarised in Table 1.

Table 1 Teacher’s characteristics in the CLUSTer cohort.

Variables	n	Weighted 1	
Sociodemographic			
Age	14,143	40.22 ± 8.85	
Sex	14,144		
Male		17.0 (16.0, 17.0)	
Female		83.0 (83.0, 84.0)	
Ethnic	14,144		
Malay		77.0 (76.0, 78.0)	
Chinese		14.0 (13.0, 15.0)	
Indian		8.0 (7.4, 8.7)	
Other races		0.8 (0.6, 1.1)	
Education	11,820		
Secondary		3.3 (2.9, 3.7)	
Diploma		2.5 (2.2, 3.0)	
Degree		79.0 (78.0, 80.0)	
Master and above		15.0 (14.0, 16.0)	
Marital status	12,160		
Single		12.0 (11.0, 13.0)	
Married		86.0 (85.0, 87.0)	
Divorced		1.4 (1.1, 1.8)	
Widowed		1.0 (0.8, 1.3)	
Family history of DM	11,834	51.0 (50.0, 53.0)	
Lifestyles			
Fruit consumption (servings/day)	8,971	0.87 ± 1.01	
Vege consumption (servings/day)	8,517	1.48 ± 1.37	
Fruit and vegetable consumption (adequate)	8,345	2.3 (2.0, 2.6)	
Sleep hours (weekday)	11,214	5.59 ± 1.00	
Sleep hours (weekend)	11,261	6.65 ± 1.13	
Physical activity (METs-minutes/week)	14,144	1344.58 ± 2908.50	
Duration of sitting (minutes)	5,637	191.67 ± 148.56	
Smoking status	11,724	4.0 (3.5, 4.5)	
Alcohol consumption	11,097	3.7 (3.3, 4.1)	
Waist circumference (cm)	13,548	81.52 ± 11.31	
Abdominal obese (by sex)	13,548	52.0 (51.0, 53.0)	
Depression score	11,816	6.22 ± 6.22	
Anxiety score	11,844	8.33 ± 6.45	
Stress score	11,685	9.80 ± 6.90	
Outcomes			
Type 2 diabetes mellitus	14,144		
Known T2DM		4.1 (3.7, 4.5)	
Undiagnosed T2DM		5.4 (4.9, 6.0)	
Impaired fasting blood glucose (IFG)	12,757		
Yes		5.6 (5.1,6.2)	
Notes.

SD Standard deviation

CI Confidence Interval

1 mean ± SD or % (95% CI)

A comparison between the prevalence of known T2DM, undiagnosed DM, and IFG between teachers and the general population from the NHMS 2015 is shown in Fig. 1. The comparison based on age ranged from 25 to 59 years to match the general service age of teachers in schools. Comparatively, the weighted prevalence of known and undiagnosed T2DM among teachers was lower than that of the general population. However, when comparing these two diseases status among teachers, the prevalence of undiagnosed T2DM was higher than that of known T2DM across the observed age groups, particularly from age group 35 onwards. In addition , the prevalence of IFG among teachers was higher than that of the general population, especially among those aged 45 to 59 years.

Figure 1 Weighted prevalence on known T2DM, undiagnosed DM, and IFG among teachers in the CLUSter cohort: A comparison with the general population in the NHMS 2015.

The selected age range was based on the period during which teachers served in school.

In the inferential analysis, 11,412 out of 14,144 teachers were included, excluding those who had not answered the majority of the survey questionnaires.

Lifestyle factors associated with T2DM and IFG among teachers

Bivariate analyses of factors associated with T2DM and IFG levels are shown in Table S2. Factors associated with T2DM and IFG, from bivariate to multivariable regression plots, are described in separate subsections.

T2DM and its associated lifestyle factors

Of all variables included in the bivariate analyses associated with T2DM, all sociodemographic variables including age, sex, ethnicity, education level, marital status, and family history of T2DM were found to be significant at p value <0.25. The significant lifestyle variables were waist circumference, sitting duration, sleep duration, depression, anxiety, and stress, with p values at the 0.25 level. No multicollinearity issue (variance inflation factor <5) was found among the 12 variables included in the multivariable logistic model.

Multivariable analysis showed that, the only lifestyle variable that remained significantly associated (p value <0.05) with T2DM was waist circumference after adjusting for all covariates. This finding concurs with the literature indicating that waist circumference is indeed a good indicator to predict the risk of developing T2DM (Siren, Eriksson & Vanhanen, 2012). Additionally, an ad hoc interaction analysis indicated that there was a significant interaction between age, history of T2DM, and waist circumference. The final model showed that every one unit increase in waist circumference increased the odds of acquiring T2DM by 14%, with 95% CIs from 1.08 to 1.20, p value <0.001 (Table S3).

Two multivariable regressions were plotted to display the interaction effect of age and family history of T2DM on waist circumference in relation to T2DM among teachers (Fig. 2). From the left figure, denoted as A, the risk of developing T2DM differed with waist circumference (70 cm onwards) by age group. However, the risk of T2DM increased exponentially when waist circumference exceeded 80 cm, especially among those aged 45 and 55 years old. Teachers whose waist circumference exceeded 110 cm had an approximately equal risk of developing T2DM regardless of their age. When further stratified by family history of T2DM, there was an exponential increment in T2DM risk as teachers’ waist circumference increased, where those with a family history of T2DM had a consistently higher risk than their counterparts. However, when the waist circumference exceeded 100 cm, there was no obvious difference in the risk of acquiring T2DM regardless of the family history of the disease.

Figure 2 Multivariable regression plots on waist circumference in relation to predicted percentage on T2DM.

(A) By age; (B) by family history of T2DM. Dotted black line indicates overlapping confidence intervals between comparison groups beyond this point (100 cm).

Impaired fasting glucose (IFG) and its associated lifestyle factors

In bivariate analysis (Table S2), covariates such as age, sex, ethnicity, marital status, and family history of T2DM were significantly associated with IFG. All lifestyle factors were found to be significant at p value <0.25 level related to the outcome, with the exception of sitting duration and acute depression and anxiety levels. Multicollinearity was not observed for any of the 12 variables included in the multivariable model.

Multivariable logistic regression (Table S3) showed that only waist circumference and physical activity were significantly (p value <0.05) associated with IFG after accounting for the effect of other covariates. Hence, the final adjusted odds ratio of waist circumference and physical activity was as follows: One centimetre increase in waist circumference increased the odds of IFG by 10.0%, with a 95% confidence interval of 1.05 to 1.15. Conversely, teachers with moderate (aOR = 0.71, 95% CI: 0.52, 0.98) or high (aOR = 0.56, 95% CI: 0.40, 0.80) physical activity were protective against IFG compared to those with low physical activity.

A multivariable regression plot on waist circumference and physical activity in relation to IFG is illustrated in Fig. 3. Three significant interaction terms, namely age with waist circumference, ethnicity with waist circumference, and ethnicity with physical activity, were included. Figure 3A shows the effect of age on waist circumference plotted against the risk of IFG. Similar to T2DM, IFG risk increased exponentially with waist circumference ranging from 60 cm to 120 cm, where older teachers tended to have the highest risk across all waist circumferences. However, the risk was approximately the same when waist circumference was approximately 100 cm.

Figure 3 Multivariable regression plots on waist circumference and physical activity associated with impaired fasting glucose (IFG) status among teachers in the CLUSTer cohort.

(A) By age; (B) by ethnicity; (C) by ethnicity.

When observing waist circumference by ethnicity (Fig. 3B), there was a trend of increasing risk among Malay and Chinese teachers when their waist circumference increased. There was no exponential increase among Indian teachers in relation to waist circumference, even though their risk was the highest even with a normal waist circumference below 80 cm. A large confidence interval for Indian teachers indicates a low sample size. Hence, these results must be interpreted carefully.

On the other hand, the plot of waist circumference by race against the outcome in Fig. 3C shows that there was a decreasing risk of impaired glucose from low to high physical activity. This trend is consistent for all races except Chinese. Moderately active Chinese teachers tended to have a slightly higher point estimate of the risk of acquiring impaired glucose than low- and high-active teachers, but the confidence intervals for all physical activity groups among Chinese teachers overlapped. This indicated that the effect of physical activity on the risk of acquiring IFG was minimal among Chinese individuals compared to other races.

When stratified by gender, most sociodemographic, lifestyles, mental health status variables were comparable between male and female teachers. The average physical activity of the male teachers was slightly higher than that of the female teachers. Male teachers tended to smoke and consume more alcohol than their female counterparts. As for the outcomes (known T2DM, undiagnosed T2DM, and IFG), male teachers recorded higher percentages of all these outcomes compared to female teachers. All the details of the weighted descriptive analysis are shown in Table S4. Additional regression models stratified by gender showed that the plots were similar (Figs. S2 & S3). This indicates that the stratified analysis by gender did not provide additional information, probably due to the imbalance in the numbers of male and female teachers in the CLUSTer cohort.

Discussion

To the best of our knowledge, this is the first study to report the prevalence of T2DM, undiagnosed DM and IFG, among teachers in Malaysia. When comparing the findings within teachers across countries, the prevalence of known T2DM among Malaysian teachers (4.1%) was lower than that of teachers in other countries such as India (6.5%) and Nigeria (5.6%), with matched teachers’ age ranges (Elegbede, Sanni & Alabi, 2022; Llone et al., 2014).

Malaysia is a multi-racial country with Malay as the dominant race, followed by Chinese, Indians, and other minorities. Ethnic diversity within unique cultures results in different lifestyle behaviours (Teh, Tey & Ng, 2014). Ethnic disparities in glycaemic control may be attributed to differences in insulin sensitivity between races (Chew et al., 2011; Tan et al., 2015). For instance, Chinese and Malays were found to be more insulin-sensitive, so they were less prone to co-living with higher blood glucose levels than Indians. Hence, ethnic variation among teachers in Malaysia may explain the lower prevalence of T2DM compared to that in India and Nigeria, which are dominated by a single ethnic group.

The prevalence of undiagnosed DM was found to be comparable to that of known T2DM among teachers, however, IFG among teachers was surprisingly higher than that in the general population at older ages. These scenarios are likely related to the teachers’ job entity. Teachers have a demanding job in which they need to be multi-taskers in school, handling both teaching and administrative work. Thus, long-term exposure to heavy workloads, notably mental workloads, can elevate the blood glucose levels. In addition, mental workload can have a synergistic effect with age (Kusnanto et al., 2020). As teachers age, metabolic changes result in glucose dysregulation (Chia, Egan & Ferrucci, 2018). This scenario, when coupled with heavy workloads, causes teachers’ stress levels to increase, leading to an increase in cortisol and blood glucose levels. This explains why Malaysian teachers aged 40 years and above were more likely to have T2DM and IFG than younger teachers. Apart from being overwhelmed by workloads, inadequate knowledge, or awareness of diabetes can also affect the tendency to acquire the disease. Although there have been no studies on knowledge and awareness among Malaysian teachers of non-communicable diseases, there is evidence that teachers have low to moderate knowledge of T2DM elsewhere (Almehmad et al., 2018; Aycan et al., 2012).

Of all the lifestyle factors included, only waist circumference was associated with the risk of T2DM. Waist circumference and physical inactivity were related to IFG. The current finding on waist circumference, as a proxy for abdominal obesity, is associated with T2DM and IFG among teachers, in concordance with other studies (Jiang et al., 2019; Lee, Hairi & Moy, 2017; Narayanappa, Manjunath & Kulkarni, 2016). Abdominal obesity is characterised by the accumulation of fat in the abdominal region, and this chronic condition is considered one of the most prominent risk factors for type 2 diabetes (Siren, Eriksson & Vanhanen, 2012). The aetiology of obesity in T2DM is related to disturbances in the cellular insulin activity. Obesity causes macrophages to release inflammatory cytokines that inhibit insulin sensitivity, and simultaneously enable immune cells to reside within adipose tissue, resulting in inflammation in these tissues, thereby promoting insulin resistance (Zatterale et al., 2020). In addition to macrophage cytokine release, obesity increases the uptake of non-esterified fatty acids without beta-oxidation, leading to excessive lipid metabolites in the body. This can ultimately disrupt the insulin signalling pathway (Papaetis, Papakyriakou & Panagiotou, 2015). Therefore, obesity can be an intermediate outcome where it synergizes well with poor lifestyles in relation to downstream outcomes such as poor glucose control and T2DM.

Waist circumference was used in this study (instead of body mass index, BMI) to proxy the obesity status among teachers. Waist circumference is a more accurate measure of abdominal obesity (Jeon, Jung & Jee, 2019), while BMI estimates general obesity. A sensitivity analysis through a comparison of models (with and without the inclusion of BMI) revealed a minor improvement in the Akaike Information Criterion (AIC) (Table S5). This indicates that the combination of waist circumference and BMI only improve diabetes prediction among teachers slightly, probably due to a strong correlation (r = 0.75, p = <0.001) between the BMI and waist circumference. Including two highly correlated variables in a regression model may affect the risk prediction on the disease. In addition, the relatively high correlation between waist circumference and BMI suggested that teachers with high BMI are also likely to have large waist circumference. Besides, it is noteworthy that when the study group (CLUSTer) is predominately females, waist circumference is a more sensitive anthropometric measure to link with T2DM (Hajian-Tilaki & Heidari, 2015).

The current findings also showed that physically active teachers had fewer problems regulating their blood glucose levels, which is in line with Damtie et al. (2021), although they investigated hypertension among teachers. Nevertheless, all metabolic disorders such as diabetes, hypertension, and dyslipidaemia share similar lifestyle factors. Physical activity has short- and long-term effects on glucose levels. Short-term effects of physical activity can control blood glucose levels, whereas long-term effects of physical activity can enhance insulin action, blood glucose control, and fat oxidation and storage within the muscle (Colberg et al., 2010). In short, physical activity increases skeletal muscle mass, which in turn increases the body’s demand for glucose. This condition balances the glucose produced by hepatic cells. Physical activity also plays an important role in regulating body fats, thereby reducing the risk of a physically active person becoming obese.

Lifestyle factors such as fruit and vegetable consumption, smoking status, and alcohol consumption were not associated with T2DM and IFG among our teachers. One study stated that Malaysians had inadequate daily consumption of fruits and vegetables (Nur Shahida, Norzawati & Faizah, 2015), which is similar to our findings. Inadequate vegetable consumption among teachers may be due to the cooking styles of various ethnicities, or a lack of awareness about updates on recommended fruit and vegetable serving sizes from the Malaysian food pyramid (Izzah et al., 2012). In addition, the proportion of teachers who smoke and drink alcohol was also relatively low, as these substances are not common among teachers and both products are strictly prohibited in school compounds. Sleep and acute mental health conditions were also not found to be related to the outcome of interest among teachers in the current study, despite these two lifestyle factors deemed closely related to teachers’ well-being (Musa, Moy & Wong, 2018; Pau et al., 2022; Tai, Ng & Lim, 2019). These lifestyle factors were also closely associated with waist circumference in the ad hoc analysis (Table S6), suggesting that these factors exacerbated impaired glucose tolerance and ultimately T2DM, rather than being a direct cause (Barone & Menna-Barreto, 2011; Wong et al., 2019). Sleep deficiency, acute stress, and other mental health conditions are often associated with stimulation of appetite (Hirotsu, Tufik & Andersen, 2015). Affected individuals usually consume additional food, leading to obesity, a condition that can potentially disrupt insulin sensitivity. Therefore, although sleep and acute mental health status did not show any significant association with the outcomes in the current study, these lifestyle factors should be included in statistical models for predicting T2DM events.

To the best of our knowledge, this study had the largest sample size among Malaysian teachers. In addition, this study incorporated an interaction term into the multivariable analysis. This allowed the effects of common effect modifiers on lifestyle indicators of interest to be observed, thereby providing more accurate disease modelling. However, this study has several limitations. First, this was a cross-sectional study, in which the causality of lifestyle indicators with disease outcomes could not be established. Second, as a gender and ethnically imbalanced occupation, the findings from this study may not be extrapolated to other teaching professions such as lecturers in Malaysia. Third, social desirability bias among teachers may occur, especially when they answer high risk behaviour questionnaires such as smoking, alcohol consumption, and mental health status.

This study successfully identified relevant lifestyle components associated with T2DM, setting the framework for future research. Although the interactions between lifestyles seem complex in relation to T2DM, there is a promising avenue for progress by incorporating a newly developed composite score. This composite score has the potential to improve disease prediction among teachers. In addition, a follow-up event for teachers needs to be conducted so that researchers can utilise the composite score to predict the time to develop T2DM among teachers, particularly among those with unhealthy lifestyles.

Conclusions

The prevalence of T2DM among teachers was lower than that in the general population, however the proportion of undiagnosed DM and IFG levels among older teachers was relatively high. Monitoring potential lifestyle indicators, such as waist circumference and physical activity, could modify the risk of developing T2DM. Understanding the relationship between these lifestyle factors and identifying the core lifestyle risk factors associated with T2DM could help reduce future healthcare costs and increase teachers’ productivity.

Supplemental Information

Figure S1 Statistical weights calculation for teachers in the CLUSTer cohort

Click here for additional data file.

Figure S2 A comparison of regression models on lifestyle factor associated with T2DM

(A & D = Both male and female teachers), (B & E = Male teachers), (C & F = Female teachers)

Click here for additional data file.

Figure S3 A comparison of regression models on lifestyle factor associated with IFG

(A, D & G = Both male and female teachers), (B, E & H = Male teachers), (C, F & I = Female teachers). Noted that interaction terms were remove for regression model on male teachers due to errors while executing the R codes.

Click here for additional data file.

Table S1 Operational definitions on variables

Click here for additional data file.

Table S2 Bivariate analysis on factors associated with T2DM and impaired fasting blood glucose (IFG) among teachers

Click here for additional data file.

Table S3 Multivariable analysis on lifestyle factors associated with T2DM and impaired fasting glucose (IFG) among teachers

Click here for additional data file.

Table S4 Weighted descriptive statistical analysis on variables by gender

Click here for additional data file.

Table S5 Regression models’ performance on T2DM and IFG prediction: A comparison between models

Click here for additional data file.

Table S6 Lifestyle factors associated with waist circumference among teachers in CLUSTer cohort

Click here for additional data file.

Supplemental Information 10 R codes for all data analyses

Click here for additional data file.

Data S1 Dataset for a comparison of T2DM and IFG among teachers in the CLUSTer cohort with the Malaysia National Health and Morbidity Survey

Click here for additional data file.

Data S2 Dataset for complex descriptive analysis of teachers characteristics

Click here for additional data file.

Data S3 Dataset for bivariate and multivariable analyses on lifestyles and other covariates associated with T2DM and IFG among teachers in the CLUSTer cohort

Click here for additional data file.

Supplemental Information 14 BMI dataset for sensitivity analysis (cross-check with waist circumference in predicting T2DM and IFG risks among teachers)

Click here for additional data file.

Supplemental Information 15 STROBE checklist

Click here for additional data file.

Additional Information and Declarations

Competing Interests

Author Contributions

Human Ethics

Data Availability

The authors declare there are no competing interests.

Yit Han Ng conceived and designed the experiments, performed the experiments, analyzed the data, prepared figures and/or tables, authored or reviewed drafts of the article, and approved the final draft.

Foong Ming Moy conceived and designed the experiments, authored or reviewed drafts of the article, and approved the final draft.

Noran Naqiah Hairi conceived and designed the experiments, authored or reviewed drafts of the article, and approved the final draft.

Awang Bulgiba conceived and designed the experiments, authored or reviewed drafts of the article, and approved the final draft.

The following information was supplied relating to ethical approvals (i.e., approving body and any reference numbers):

University of Malaya Medical Centre Medical Research Ethic Committee granted Ethnical approval to carry out the study (UMMC MREC ID: 950.1).

The following information was supplied regarding data availability:

The raw data is available in the Supplemental Files.

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
