# Peer review of "Prevalence of type 2 diabetes mellitus and impaired fasting glucose, and their associated lifestyle factors among teachers in the CLUSTer cohort"

_PeerJ, doi:10.7717/peerj.16778_

## Round 0.1 · original submission · Major Revisions

The authors should address all the issues raised by the reviewers.

·

Basic reporting

The article meets the journal’s guidelines. Ethical approval statements have been checked.
The data has been deidentified and the experiments conducted have been ethical.
The supplemental files (figures and datasets), and the figures and tables in the manuscript have been checked. The R script has been successfully reproduced on a windows 10 machine with Rstudio. The sample sizes have been confirmed as well.

Overall, I commend the authors that the research layout is clear to understand. Professional English language has been used throughout. Figures are relevant and labelled.

Experimental design

Revisions
• Please mention the expansions along with the abbreviations at the first occurrence in the article (eg. Impaired fasting glucose (IFG))
• Line 61: The prevalence of T2DM in the general population increased but by what amount, please mention. Perhaps similar studies on NHANES data (eg. doi: 10.1001/jama.2021.9883) could help.
• Line 62-64: Sentence starting with Concurrently needs a grammar check, rephrase it please.
• Line 64: Please mention, if available, by what amount the IFG levels have increased.
• Great job noticing the fact that there is a lack of data on non-communicable diseases among the teacher population in Malaysia.
• Study sampling design has been clearly explained along with the study variables.
• Line 117-119: The idea leading up to multiplying the teacher’s weight and the school’s weight needs a bit more explanation. Perhaps authors can use Sharon Lohr’s book on sampling design which explains the complex analysis in detail to address this point.
• Line 128: p value < 0.25 instead of “p < 0.25”.
• Line 131: Variance inflation factors < 5 suggesting no multicollinearity is sort of a thumb rule, I have seen studies where the threshold is 10. Please provide a reference if possible on the threshold for multicollinearity. Maybe the authors can find John Fox’s original paper on multicollinearity to cite here.

Validity of the findings

• Line 193-194: BMI (waist measurement) has already been proven to be a good indicator of T2DM https://bmcpublichealth.biomedcentral.com/articles/10.1186/1471-2458-12-631. Maybe authors could cite any of these reports to corroborate their findings.
• Line 198: Please mention the p value. It strengthens the argument.
• Line 220: p value < 0.05 instead of “p < 0.05”.
• Line 319: “Sleep deficiency, acute stress, and other mental health conditions are often associated with stimulation of appetite”, please cite the source.
• Over all the Discussion section is pretty robust.

Reviewer 2 ·

Basic reporting

The study is well thought, however, minor English corrections are required. The background provides details on the importance of the study question. The tables and figures do provide relevant information.

Experimental design

The research question is important as T2DM rates are increasing and the topic is understudied in the mentioned location. However as mentioned in the limitations, the results cannot be generalized and some of the findings are commonly observed results. Include the definitions for the main outcome under the 'Study variables' section. How was T2DM measured - Was it self reported in the 'health status' section of the questionnaire. If so, add few details under the 'Study variables'. For the 'general population', the participant characteristics are not clear. As the results are compared with the general population, adding more details will make the paper stronger. If any other dietary factors were also studied, include the details. Similar to the waist circumference interaction with ethnicity, were any interactions also checked with diet and ethnicity.

Validity of the findings

As described in the manuscript, a detailed analysis has been performed. However the study design restricts the scope of the study. The study aimed at looking at different factors associated with T2DM and the conclusion match.

Additional comments

The abstract results section - last line is vague and does not add any information. In the background section, a statement has been mentioned regarding the work performance among teachers and T2DM. If studies exist demonstrating this association, provide some literature in the manuscript background.

Reviewer 3 ·

Basic reporting

This paper is written in English, but in my opinion it needs some corrections, however I am not native speaker.
The introduction is very superficial, e.g. there is no reference to research in Europe
In row 55-56 Authors wrote: „While chronic diseases such as T2DM can also affect teachers, information on its prevalence is scarce as this non-communicable disease usually develops silently until it becomes symptomatic”. I think that it is no right explanation. Probably information about T2DM in teachers is scarse, because it is rarely studied.
Mental health conditions are not a lifestyle factors (rows 86-87), this phrase should be corrected.
When using the abbreviations T2DM, IFG for the first time, they should be explained
The structure of the article is standard

Experimental design

1. In my opinion, the procedure for selecting the study group should be briefly presented in this manuscript, even though there was a literature reference to previous studies within the CLUSTer Cohort, especially since the data differ, e.g. in the cited publication there are 6 states, and in the current one there are 5.
2. The description of the methods lacks information on what tests/questionnaires were used to assess diet and physical activity, as well asd depression, anxiety, and stress
3. There is also no data on the adopted dietary guidelines or recommended amounts of vegetables and fruit. There is also no information on the recommended level of physical activity.
4. There is also no information on how the material for the analysis of fasting glucose was collected and stored, who did it, where the analyses were carried out, etc.
5. Why did the Authors only analyse waist circumference and not include BMI. In several papers obesity or overweight estimated by BMI, is indicated as a very important risk factor for T2DM
6. How did the authors calculate the prevalence of undiagnosed T2DM. Did they assess based on their own research? If this is the result of their own tests, it should be remembered that a diagnosis of T2DM cannot be made based on a single measurement (Line 163).
7. In my opinion, it should be stated how many women and men there were in the group and all data in Table 1 should be divided by gender. There is a lot of data on differences in prevalence of T2DM between men and women (Shepard BD. Sex differences in diabetes and kidney disease: mechanisms and consequences. Am J Physiol Renal Physiol. 2019 Aug 1;317(2):F456-F462. doi: 10.1152/ajprenal.00249.2019, Carlsson S, Andersson T, Talbäck M, Feychting M. Incidence and prevalence of type 2 diabetes by occupation: results from all Swedish employees. Diabetologia. 2020 Jan;63(1):95-103. doi: 10.1007/s00125-019-04997-5).
8. It would be worth conducting all analyzes separately for men and women

Validity of the findings

Results are important but not entirely justified, because the description of the methods lacks information on what tests/questionnaires were used to assess diet, physical activity, depression, anxiety, and stress as well as the adopted dietary and physical activity guidelines.

Additional comments

no comment

---

## Round 0.2 · Minor Revisions

The reviewers are quite satisfied about the changes made. Please address the last request from Reviewer 3.

·

Basic reporting

The authors have addressed all the suggested revisions. The article is ready for publishing in my opinion.

Experimental design

no comment

Validity of the findings

no comment

Reviewer 2 ·

Basic reporting

na

Experimental design

na

Validity of the findings

na

Reviewer 3 ·

Basic reporting

This manuscript after Author’s corrections and clarifications is much more comprehensive and meets the standards of a scientific paper. The introduction is rather unchanged, but the Authors took into account the comment regarding the lack of reference to the situation in Europe.

Experimental design

The Authors included information about the methods/questionnaires used (diet, physical activity, depression, anxiety and stress) and adopted guidelines and recomendations. However, I do not fully agree with the Authors' answer regarding the advantage of measuring waist circumference over BMI, as both parameters are usually used. I understand that the Authors probably do not have data on BMI. But in the content of the work they should explain why they used only waist circumference

Validity of the findings

The results are important and now much more documented than in the previous verssion.

Additional comments

Dear Authors, thank you for your explanations and corrections. I accept them, but suggest to explain in the manuscript the reasons why BMI was not calculated.

---

## Round 0.3 · Minor Revisions

Please, improve your answer to the remaining concern of Reviewer 3.

Reviewer 3 ·

Basic reporting

Dear Authors,
thank you for your answer and explanation. I understand that you didn’t calculate the BMI and maybe you even haven’t data for it. But according to a Consensus Statement from the IAS and ICCR Working Group on Visceral Obesity, it is recommended to evaluate both („The combination of BMI and waist circumference identifies a high-risk obesity phenotype better than either measure alone”) [Ross R, Neeland IJ, Yamashita S, Shai I, Seidell J, Magni P, Santos RD, Arsenault B, Cuevas A, Hu FB, Griffin BA, Zambon A, Barter P, Fruchart JC, Eckel RH, Matsuzawa Y, Després JP. Waist circumference as a vital sign in clinical practice: a Consensus Statement from the IAS and ICCR Working Group on Visceral Obesity. Nat Rev Endocrinol. 2020 Mar;16(3):177-189. doi: 10.1038/s41574-019-0310-7]. Therefore please disscuss this problem more thoroughtly in your paper.

Experimental design

See Review 1 and 2

Validity of the findings

See Review 1 and 2

Additional comments

I have no additional comments

---

## Round 0.4 · accepted · Accept

The authors have satisfactorily addressed all the reviewers' comments so that the manuscript is now ready for publication.

Reviewer 3 ·

Basic reporting

Thank you for additional changes made in the manuscript. I haven't more remarks.

Experimental design

Now is o.k.

Validity of the findings

This paper is important, because there is not much data on the prevalence of diabetes among teachers

Additional comments

All comments I presented in my previous reviews